# Spatial Patterns and Determinants of Endemic Taxa Richness in the Genus *Viburnum* (Adoxaceae) in China

**Wenjun Lyu** [1,2,†] , **Shenglan Du** [3,†] , **Jiali Ying** [1,2,4] , **Veronicah Mutele Ngumbau** [5,6] , **Sheng Huang** [3,7] , **Shengwei Wang** [1,2,5,*] and **Hongtao Liu** [1,2,4,*]

1   National Germplasm Repository of Viburnum, Wuhan Botanical Garden, Chinese Academy of Sciences, Wuhan 430074, China
2   Center of Conservation Biology, Core Botanical Gardens, Chinese Academy of Sciences, Wuhan 430074, China
3   College of Forestry and Horticulture, Hubei Minzu University, Enshi 445000, China
4   University of Chinese Academy of Sciences, Beijing 100049, China
5   Sino-Africa Joint Research Center, Chinese Academy of Sciences, Wuhan 430074, China
6   East African Herbarium, National Museums of Kenya, P.O. Box 45166, Nairobi 00100, Kenya
7   Enshi Dongsheng Plant Development Co., Ltd., Enshi 445000, China
*   Correspondence: wangshengwei@wbgcas.cn (S.W.); lhongtao@wbgcas.cn (H.L.)
†   These authors contributed equally to this work.

**Abstract:** Understanding the distribution patterns and formation mechanisms of endemic taxa is essential for effective biodiversity conservation. China is an important distribution and endemic center for genus *Viburnum* in Asia. However, the distribution pattern and formation mechanism of endemic taxa of *Viburnum* remains unclear in China. In this study, we determined the distribution information of 61 endemic taxa of *Viburnum* through specimens' review and field surveys. Species distribution models were used to clarify the distribution patterns of the endemic taxa of *Viburnum*. The findings shows that the hotspot for overall endemic taxa of *Viburnum* in China is mainly distributed in temperate and subtropical mountainous areas, and the highest richness in the mountainous regions were around the Yunnan-Guizhou Plateau and the Sichuan Basin. About one-third of the endemic taxa of *Viburnum* were rare species, whose distribution area was scattered and lacked protection. The distribution pattern of the endemic taxa of genus *Viburnum* can be well explained within the three hypotheses of environmental energy, water availability and climate seasonality. This study provides additional understanding and explanation of endemic species richness distribution and their formation mechanisms. In addition, it provides conservation measures for endemic taxa of genus *Viburnum* to guide conservationists and policy makers in China.

**Keywords:** endemic taxa; *Viburnum*; environmental factors; geographical pattern; conservation

## 1. Introduction

The heterogeneous distribution patterns of biodiversity and its formation mechanisms have long been a research topic among ecologists and conservationists [1,2]. Species richness is an important part of biodiversity and understanding its geographic patterns can also provide important information for spatial planning conservation, and sustainable use of biodiversity resources [3,4]. Meanwhile, species richness of endemic, rare or threatened plants play a central role in identifying the priority sites for plant conservation [5].

Over the past few decades, hundreds of macroecological and biogeographical hypotheses have been used to explain the patterns of species richness and the formation mechanisms [6–8]. In particularly, the synergistic effects between climate and habitat heterogeneity drive species richness at large scales [9–11]. The water-energy hypothesis is one of the most widely used of these hypotheses, suggesting that areas with high energy and water availability will promote high species diversity and richness [12,13]. The habitat heterogeneity hypothesis suggests that areas of high topographic complexity can

promote species differentiation through geographical isolation [4,14]. Furthermore, climatic seasonality (severe winters) and climate change since the Quaternary can also influence species richness patterns [4,15]. Recent studies suggest that species richness patterns, particularly those of endemic and rare species with low dispersal ability and narrow ranges, may be affected by the combined effects of interactions between contemporary climate and long-term climate stability [16,17]. Furthermore, the interpretation of species richness patterns and the relative importance of explanatory factors may vary depending on the study extents, research scale, spatial resolution, and species groups [18–21]. However, the richness patterns and explanatory hypotheses for woody plants genus with high proportion of endemism have received less attention and are not well understood.

*Viburnum* L (Adoxaceae), is a highly ornamental woody flowering genus, and popular for its horticultural values which are characterized by its showy flowers, brightly colored fruit, and attractive foliage [22–24]. The genus *Viburnum* has over 200 species worldwide, mainly distributed in the subtropical and temperate regions of Asia, South America and Europe [22,25]. It was first placed in the traditional Caprifoliaceae family and later placed in the Adoxaceae based on morphological and molecular evidence [26,27], and can be divided into 10 sections in traditional taxonomy [28]. Among those sections, the Sect. *Megalotinus*, Sect. *Solenotinus*, Sect. *Tinus*, Sect. *Tomentosa*, Sect. *Viburnum* and Sect. *Pseudotinus* were mainly distributed in Asia, the Sect. *Lentago* and Sect. *Oreinotinus* were mainly distributed in America, the Sect. *Odontotinus* was mainly distributed in both Asia and America, with the Sect. *Opulus* could be found on all continents in the circumpolar Arctic region [28]. According to the recent molecular phylogenetic findings, the global genus *Viburnum* could be broadly divided into 18 sections [29,30]. The genus *Viburnum* probably originated and initially diversified in Old World montane tropical forests, and several lineages gradually migrated to the colder forests in the higher latitudes of Asia. Subsequently, some lineages extended into Europe, and some lineages may have entered the New World through Belincia in the Eocene [30,31]. From the Oligocene onward, there was less species formation and more extinction in the lowland tropical lineages, and increased diversification in temperate lineages. After the Miocene, it spread again to tropical forests and boreal cold forests [30]. Today, the center of phylogenetic diversity is in Southeast Asia, but the greatest species diversity is found in central and southern China [32–35].

China has the largest species number of the genus *Viburnum* in the world and the distribution center of this genus in Asia [25]. According to the Flora of China and the latest research of new distribution records and new species, there are 73 species (98 taxa) of the genus *Viburnum* in China [25,36], of which 45 are endemic to China [25]. The Flora Reipublicae Popularis Sinicae describes the genus *Viburnum* is widespread in all provinces of China, with the largest number of species in the southwest [37]. However, the geographical distribution pattern and endemic centers of endemic taxa of the genus *Viburnum* in China and their driving factors are unclear, which has hampered research on the evolution and conservation of endemics of the genus *Viburnum*.

In this study, we aimed at solving the following problems: (1) what is the geographical distribution pattern of endemic taxa of *Viburnum* in China; (2) What are the possible causes of the pattern of endemic taxa of *Viburnum* in China? This study will provide a basis for research on the evolution and conservation of endemic taxa of *Viburnum* in China.

## 2. Methods

### 2.1. Plant Taxa and Distribution Data

The list of plant taxa and taxonomic information in this study was based on the Flora of China [25]. Information on new species and new distribution of *Viburnum* in China has also been recorded through literature review. The distribution data of endemic taxa of *Viburnum* in China was mainly obtained from digitized specimens and field surveys. Specimens data were obtained from the Chinese Virtual Herbarium (www.cvh.ac.cn, accessed on 10 September 2021), the National Specimen Information Infrastructure (www.nsii.org.cn, accessed on 10 September 2021), and Herbariums from other research institutes, which in-

clude Shanghai Chenshan Botanical Garden, Hangzhou Botanical Garden, Xishuangbanna Tropical Botanical Garden of the Chinese Academy of Sciences, South China Botanical Garden of the Chinese Academy of Sciences, and Wuhan Botanical Garden of the Chinese Academy of Sciences. From the digitized specimen information, we obtained latitude and longitude data. For the specimens which just had collection location name, geographical coordinates were obtained from the Chinese National Geographical Names Database (www.nfgis.nsdi.gov.cn/, accessed on 10 September 2021) and Google Earth. The field survey data was obtained from the Wild Flora Introduction Department of Wuhan Botanical Garden of the Chinese Academy of Sciences from 1980 to 2021, and from Enshi Dongsheng Plant Development Co., Ltd, Enshi, China. from 2008 to 2021. Finally, a total of 7119 valid distribution information was obtained after removing doubtful, duplicate records, ambiguous information, and misidentified specimens.

### 2.2. Species Distribution Model

In this study, Maximum entropy model (MaxEnt) [38] was chosen for predicting the potential range of endemic species of genus *Viburnum*. Maxent was specifically developed to model species distributions with presence-only data, and previous studies have shown that it performs well with few distributed data and has been less affected when positional errors occur [39–41]. In this study, Maxent was run according to the following modelling rules: linear features and quadratic features for ≥5 and <15 collection records, and adding hinge features for ≥15 records [40,42]. The random test percentage was 25% with 100 replicates. The prediction accuracy of the model was evaluated by the ROC curve. The ROC curve is a methodological technique, and the AUC value can qualitatively assess the prediction accuracy [43,44].

In this case, 19 bioclimatic variables (the average for the years 1970–2000) from the WorldClim (https://www.worldclim.org/, accessed on 10 September 2021) dataset [45], and elevation (www.diva-gis.org, accessed on 10 September 2021) with a spatial resolution of 2.5 min (~5 km) were selected as environmental predictors for the species distribution model in this study. Since multicollinearity of variables can result overfit species distribution modeling [40,42,46], in this study we performed Spearman's rank correlation tests on 19 environmental factors, and eight factors with a Spearman's correlation of less than 0.8 were selected to participate in the modeling (Table S2).

### 2.3. Environmental Variables

Four categories of environmental variables were selected for the analysis of the drivers of species richness patterns. To test the effect of energy on species richness, we selected mean annual temperature (MAT), annual range of temperature (ART) and mean temperature of the coldest quarter (MTC). To test the effect of water on species richness, we selected mean annual precipitation (MAP) and mean precipitation of the driest month (PDM). To test the effect of climatic seasonality on species richness, we selected precipitation seasonality (PS) and temperature seasonality (TS). To test the effect of heterogeneity on species richness, we choose the range in elevation of each grid cell (maximum minus minimum elevation, REL, www.earthenv.org, accessed on 10 September 2021), terrain ruggedness index (TRI; http://www.earthenv.org/, accessed on 10 September 2021) and normalized difference vegetation index (NDVI, www.geodata.cn, accessed on 10 September 2021). We selected the climate change since the Last Glacial Maximum to track the historic climate change (MATano and MAPano), which is the absolute value of the difference in MAT and MAP between the LGM and the present [47]. Climate data were obtained from the WorldClim (http://www.worldclim.org, accessed on 10 September 2021) with the spatial resolution of 1 arc-min.

### 2.4. Data Analysis

For this study, the species richness and environmental variables were statistically analyzed in grid cells at a resolution of $50 \times 50$ km in ArcGIS 10.2. The potential distribution

area of each species was rasterized at a resolution of $50 \times 50$ km and the number of species in each grid cells was counted as species richness. Rare species were marked as species listed in the IUCN list and species with small distribution ranges, i.e., species with raster counts less than 100 (which also almost satisfied with the sample collection data points of less than 25). The environmental variables were extracted by their respective average values in each grid cells. All environmental variables and richness data were log-transformed because of their highly skewed distributions. To analyze the predictive power of these hypotheses for the endemic richness of *Viburnum*, we chose standard ordinary least squares regression (OLS) models that ignored spatial autocorrelation, and regression models that take spatial autocorrelation into account to assess the variables in these models. The two most commonly used spatial autoregressive models, the conditional autoregressive (CAR) and simultaneous autoregressive (SAR) models were chosen for this study. The three models were calculated in SAM (version 4.0, https://www.ecoevol.ufg.br/sam/, accessed on 10 September 2021) [48], along with the Akaike Information Criterion (AIC) used to rank the environmental models [8]. To assess the pattern of spatial autocorrelation in the residuals of all regression models, we plotted the correlation of Moran's I. To further explore the relative importance of environmental factors for endemic species richness in each section of *Viburnum*, we used the "calc.relimp" function in the R package "relaimpo" [49] in R 4.0.5 [50].

## 3. Results

### 3.1. Endemic Richness of Genus Viburnum in China

Through the examination of specimens and field data, we assessed 61 endemic taxa (including 38 species, 15 varieties, 5 subspecies and 3 forma) of *Viburnum* (Table S1; Figure 1), accounting for 62% of the taxa (including inner species taxa) of *Viburnum* in China. The endemic taxa belong to seven sections of genus *Viburnum* in China. Of these, Sect. *Odontotinus* is the most diverse with 21 taxa (14 species, 6 varieties and 1 subspecies), Sect. *Solenotinus* has 16 taxa (10 species, 4 varieties and 2 subspecies), Sect. Viburnum 10 taxa (6 species, 2 subspecies and 2 forma), Sect. *Megalotinus* 6 taxa (3 species and 3 varieties), Sect. *Tinus* 5 species (3 species, 1 variety and 1 forma), Sect. *Tomentosa* 2 taxa (1 species and 1 variety), and Sect. *Pseudotinus* is represented by only 1 species.

### 3.2. Geographic Patterns of Endemic Richness of Genus Viburnum in China

According to statistics of the results of the species distribution model and its distribution in $50 \times 50$ km grid cells. A total of 4162 grid cells were sampled, but only 1764 grid cells had recorded species occurrence of genus *Viburnum* hence they were used for analysis. 57.62% of the total grid cells sampled had recorded 0 endemic species. The endemic taxa richness of this genus varied considerably in China with taxa richness values ranging from 0 to 26 species per grid cell (Figure S1).

The geographic patterns of specimen collection of endemic species of the genus *Viburnum* shows a heterogeneous pattern, with its specimen collection mostly from the southern and southwestern of China (Figure 2A,B). The endemic taxa richness of genus *Viburnum* is distributed in 28 provincial administrations in China, of which Yunnan Province has the highest richness, followed by Guangxi and Sichuan Province. The northernmost distribution area is in Urumqi city (Xinjiang Uygur Autonomous Region), the southernmost area is in Ledong County (Hainan Province), the westernmost area in Shigatse City (Tibet Autonomous Region), and the easternmost area in Taipei City, Taiwan Province (Figure 2C). Based on the raster statistics of the potential distribution areas of endemic taxa, we found that the highest richness of endemic taxa of the genus *Viburnum* in the mountainous areas around the Sichuan Basin and Yunnan-Guizhou Plateau, such as the Qionglai-Wumeng Mountains, the Qinling-Daba Mountains and the Wuling Mountains regions. At the same time, the regions of Wuyi Mountains, the Nanling Mountains and the Central Mountains of Taiwan in the south-east of China also showed high richness (Figure 2D).

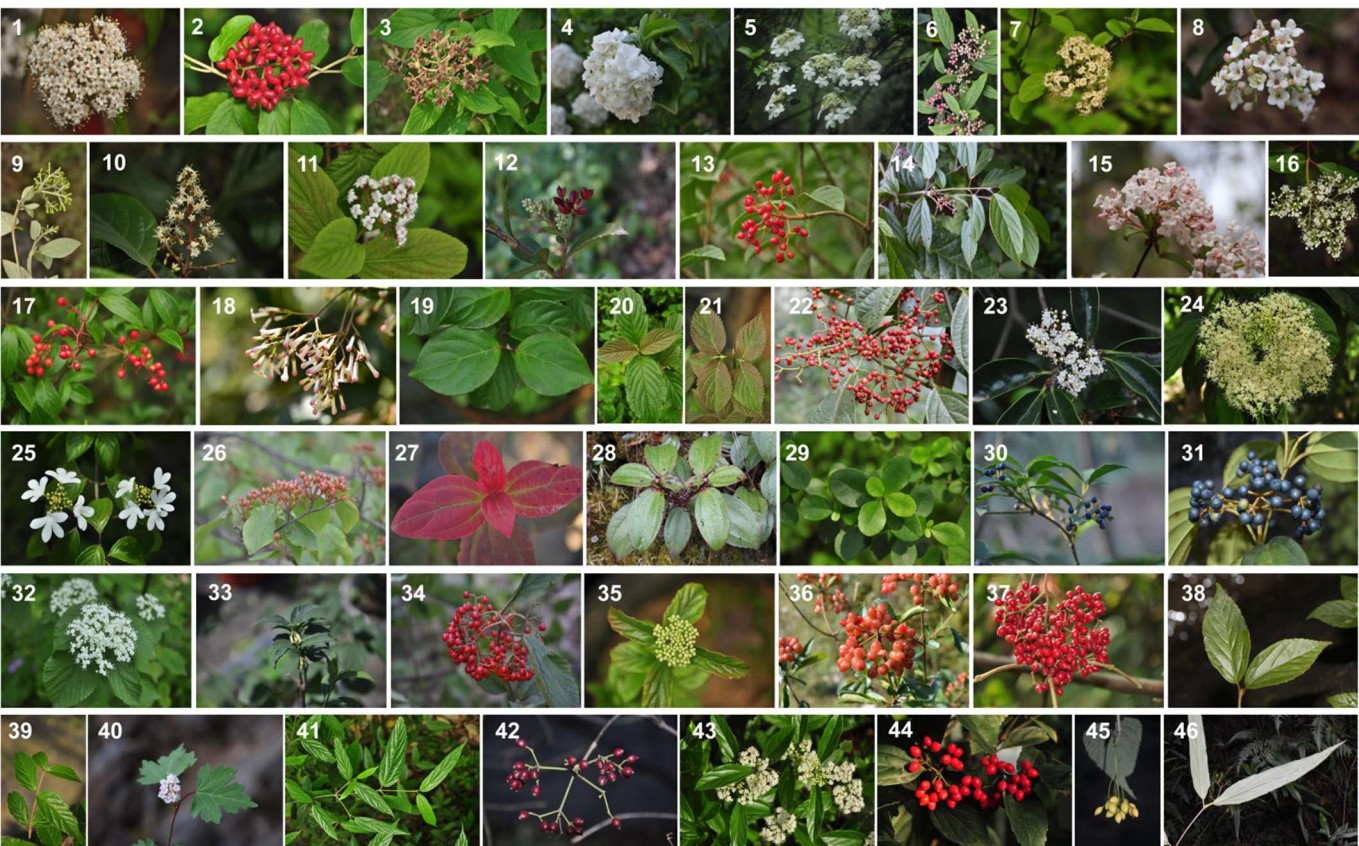

**Figure 1.** Some endemic taxa of the genus *Viburnum* in China. 1 *V. buddleifolium*, 2 *V. chinshanense*, 3 *V. glomeratum* subsp. *magnificum*, 4 *V. macrocephalum* f. *macrocephalum*, 5 *V. macrocephalum* f. *keteleeri*, 6 *V. rhytidophyllum*, 7 *V. schensianum*, 8 *V. utile*, 9 *V. congestum*, 10 *V. brachybotryum*, 11 *V. brevitubum*, 12 *V. chingii* var. *chingii*, 13 *V. corymbiflorum* subsp. *corymbiflorum*, 14 *V. corymbiflorum* subsp. *malifolium*, 15 *V. farreri*, 16 *V. henryi*, 17 *V. oliganthum*, 18 *V. taitoense*, 19 *V. tengyuehense* var. *tengyuehense*, 20 *V. trabeculosum*, 21 *V. amplifolium*, 22 *V. leiocarpum* var. *leiocarpum*, 24 *V. punctatum* var. *lepidotulum*, 24 *V. ternatum*, 25 *V. hanceanum*, 26 *V. sympodiale*, 27 *V. cinnamomifolium*, 28 *V. davidii*, 29 *V. atrocyaneum* f. *harryanum*, 30 *V. propinquum* var. *mairei*, 31 *V. triplinerve*, 32 *V. betulifolium*, 33 *V. chunii*, 34 *V. dalzielii*, 35 *V. foetidum* var. *rectangulatum*, 36 *V. foetidum* var. *ceanothoides*, 37 *V. fordiae*, 38 *V. formosanum* subsp. *leiogynum*, 39 *V. formosanum* var. *pubigeru*, 40 *V. kansuense*, 41 *V. lancifolium*, 42 *V. melanocarpum*, 43 *V. sempervirens* var. *sempervirens*, 44 *V. sempervirens* var. *trichophorum*, 45 *V. setigerum*, 46 *V. squamulosum*.

The distribution pattern of the endemic richness of the sections of genus *Viburnum* varied considerably. Of the seven sections, the Sect. *Viburnum* had a larger distribution range, with its distribution center in the area surrounding the Sichuan Basin (Figure 3A). The Sect. *Pseudotinus* (only 1 species) occurred in the areas between Sichuan and Shanghai (Figure 3B). The Sect. *Tinus* mainly occurred in south-western China, with its center of distribution in the Qionglai-Wumeng Mountains and its extensions into Guangxi Province (Figure 3C). The distribution area of Sect. *Solenotinus* was found to be widely spread, and the distribution hotspots were found to be in the southwest of China, especially in Yunnan Province and Wuling and Nanling Mountains (Figure 3D). The Sect. *Tomentosa* occurs only in the southeastern region of China, with Taiwan Province hosting all the species of this section (Figure 3E). The Sect. *Megalotinus* is mainly distributed in the southwestern and southern coastal areas, and the highest richness areas are Hainan, Taiwan, and southern Yunnan Provinces (Figure 3F). The Sect. *Odontotinus* had a wide distribution range, but the distribution center is in the south-eastern region, especially in the Wuyi and Nanling Mountains (Figure 3G). The rare species were mainly distributed in the southwest

and southeast regions, with higher richness in Yunnan and Sichuan, Hainan and Taiwan Province (Figure 3H).

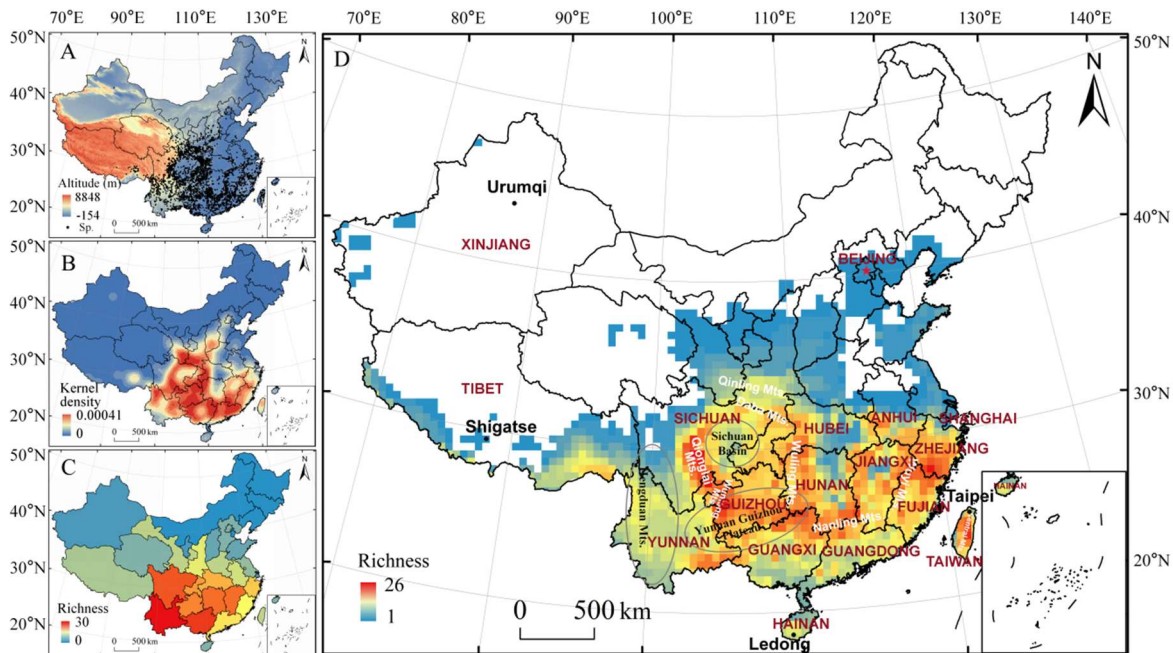

**Figure 2.** Geographical pattern of hotspots for endemic taxa of *Viburnum* in China. (**A**) Specimen Collection Patterns; (**B**) Kernel density of specimen collection; (**C**) Species richness at the provincial level; (**D**) Species richness at the 50 × 50 km grids.

### 3.3. Assessment of Variables on Species Richness

Overall, the endemic richness of genus *Viburnum* in China tends to increase with increasing longitude and decrease with increasing latitude, and areas of high richness are concentrated between 103° E and 115° E and between 25° N and 33° N (Figure S3). The endemic taxa of the genus *Viburnum* in China are mainly distributed at lower altitudes, with 50.82% of the taxa distributed at altitudes below 2000 m. Only 36.07% of the taxa are distributed at the highest altitudes in the region between 2000 m and 3000 m, while only 13.11% of the taxa have a maximum altitude above 3000 m but below 4000 m (Table S1).

The 12 explanatory variables were significant and entered the model in OLS regression analysis (Figure S3). The ART, MTC, MAP and TS were the most important variables in the model. MATano was the least important independent variable, followed by MAPano, REL, NDVI and TRI (Table 1). The OLS model explained 75.4% ($p < 0.0001$) of the variance in the transformed data (Figure 4). The Climatic seasonality hypothesis (52.9%) was the most influential factor for species richness, followed by Water availability (50.5%) and Environmental energy (49.3%) hypotheses. The least important hypothesis was Historic climate changes (16.9%). The OLS residuals showed positive spatial autocorrelation within a distance of approximately 1000 km (Figure 5). In addition, the OLS residuals were spatially clumped, showing areas of clustering with highly spatially correlated values (Figure S4).

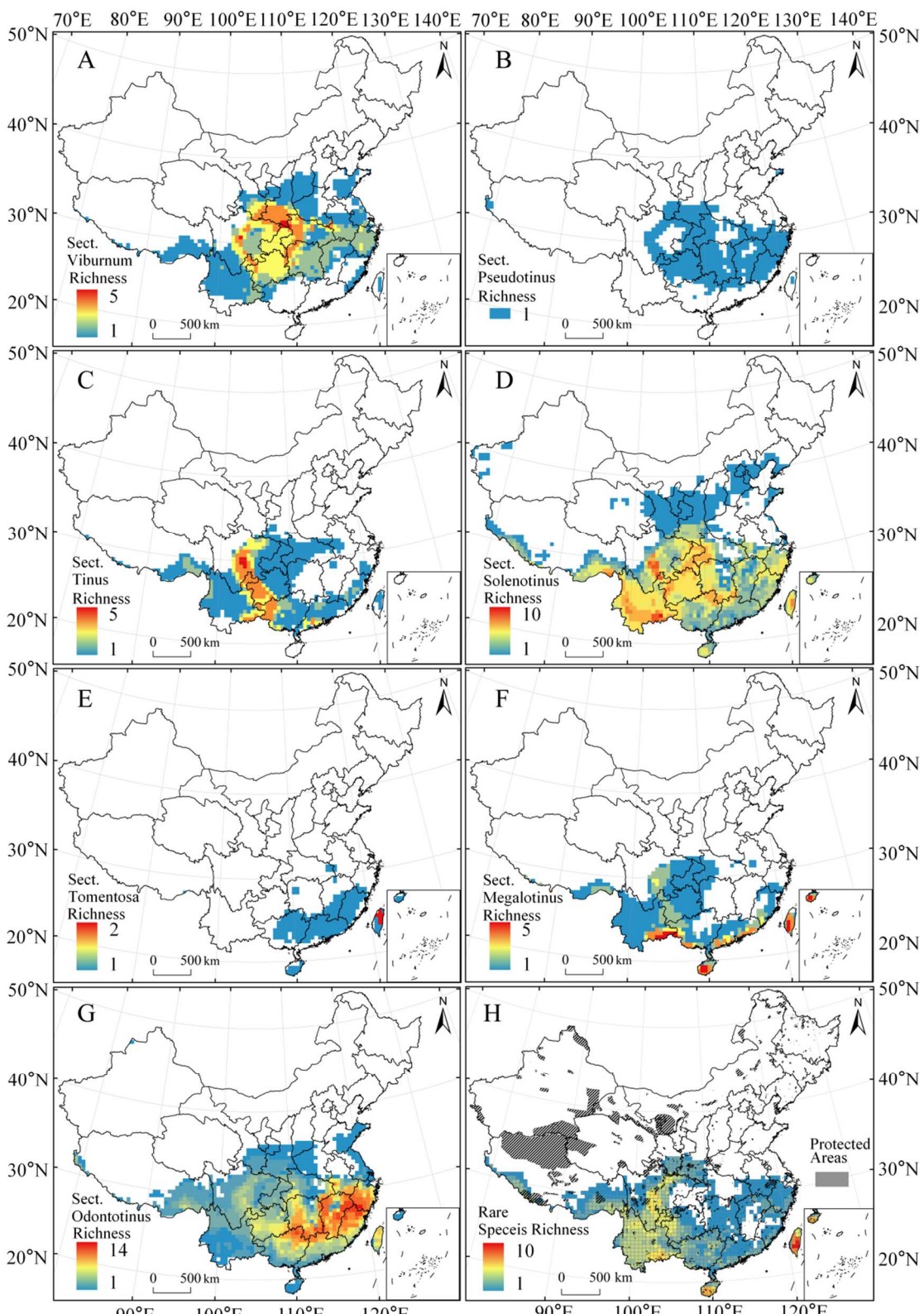

**Figure 3.** Geographical pattern of hotspots for the endemic richness of seven Sections (**A–G**) and rare taxa (**H**) of genus *Viburnum* in China.

**Table 1.** Relationships between the richness of endemic taxa of *Viburnum* and explained variables using simple ordinary least squares (OLS) model, simple conditional autoregressive (CAR) model and the simultaneous autoregressive in (SAR) model.

| | OLS Models | | | CAR Models | | | SAR Models | | |
|---|---|---|---|---|---|---|---|---|---|
| | t | $R^2$ | AIC | t | $R^2$ | AIC | t | $R^2$ | AIC |
| | Environmental energy | | | | | | | | |
| MAT | 26.498 | 0.285 | 913.599 | 21.825 | 0.300 | 875.970 | 12.282 | 0.496 | 295.932 |
| ART | −37.480 | 0.444 | 467.920 | −31.713 | 0.458 | 424.184 | −28.419 | 0.679 | −497.299 |
| MTC | 35.452 | 0.416 | 555.496 | 28.107 | 0.416 | 556.969 | 19.114 | 0.571 | 13.626 |
| | Water availability | | | | | | | | |
| MAP | 41.050 | 0.489 | 321.442 | 35.986 | 0.504 | 268.409 | 29.583 | 0.673 | −467.821 |
| PDM | 24.220 | 0.250 | 998.358 | 21.927 | 0.269 | 953.337 | 15.860 | 0.531 | 168.554 |
| | Climatic seasonality | | | | | | | | |
| TS | −26.890 | 0.291 | 898.774 | −22.896 | 0.316 | 836.727 | −24.031 | 0.639 | −289.754 |
| PS | −24.840 | 0.260 | 975.488 | −23.118 | 0.289 | 903.153 | −13.334 | 0.501 | 280.869 |
| | Heterogeneity | | | | | | | | |
| REL | 10.030 | 0.054 | 1407.314 | 13.530 | 0.109 | 1300.729 | 19.866 | 0.538 | 143.528 |
| TRI | 13.680 | 0.096 | 1327.069 | 16.096 | 0.146 | 1226.710 | 19.768 | 0.540 | 135.663 |
| NDVI | −10.880 | 0.063 | 1390.441 | −10.389 | 0.091 | 1337.344 | −5.611 | 0.444 | 468.735 |
| | Historic climate change | | | | | | | | |
| MATano | 9.466 | 0.048 | 1417.672 | 6.257 | 0.064 | 1388.166 | 4.388 | 0.447 | 462.120 |
| MAPano | 19.650 | 0.180 | 1155.614 | 15.310 | 0.168 | 1179.985 | 14.132 | 0.519 | 214.806 |

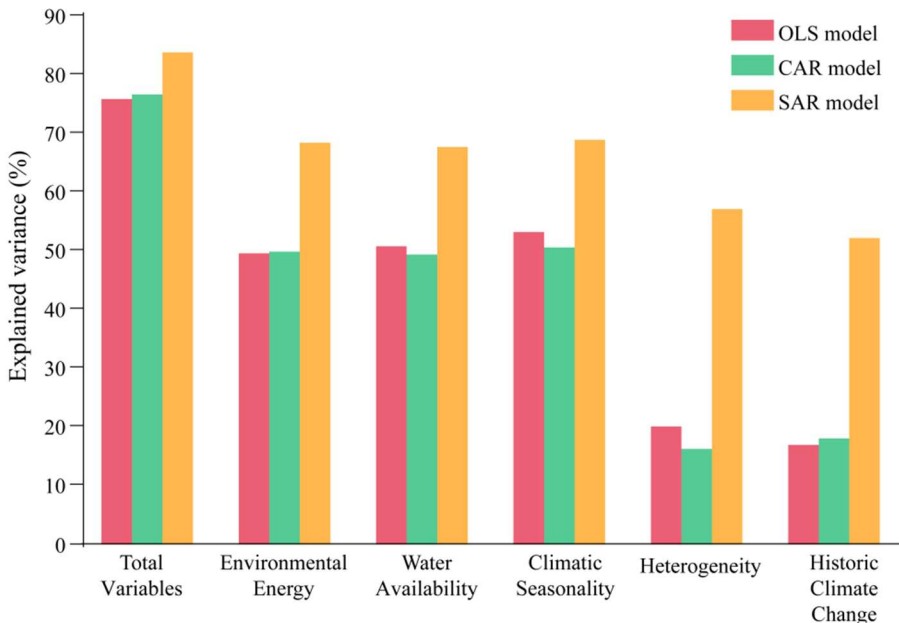

**Figure 4.** The explanation of endemic richness of genus *Viburnum* in China by different hypothetical models.

The most important explanatory factor in the CAR model was MAP, followed by ART, MTC, and TS. The explanatory rates for factors related to habitat heterogeneity and historical climate change were both low (<20%). In total, the CAR model accounted for 76.2% of the species richness pattern (Figure 3), with climate seasonality hypothesis alone explaining the highest rate (50.3%), followed by the environmental energy (49.6%) and the water availability (49.1%) hypothesis. With only 16.2 percent and 18.0 percent, respectively, habitat heterogeneity and the historical climate change hypotheses were inadequately explained (Table 1). The most important explanatory factor in the SAR model was ART,

followed by MAP, MTC, and TS. The least explained factors were NDVI and MATano (<50%). SAR models explained 83.3% of the species richness pattern (Figure 4), with the higher individual explanations being climate seasonality (68.5%) and the environmental energy hypothesis (68%), followed by the water availability hypothesis (67.3%), habitat heterogeneity (56.8%) and the historical climate change hypothesis (51.9%) (Table 1).

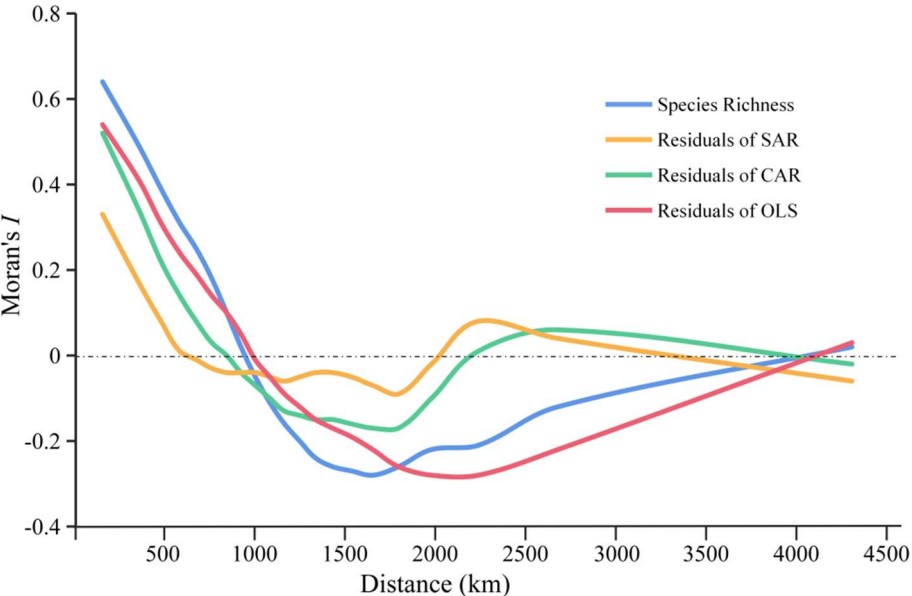

**Figure 5.** Moran's I correlogram for the residuals of OLS, CAR, and SAR models.

The residual examination of the three models shows that the OLS model has a strong spatial autocorrelation, with a maximum positive Moran's I index of 0.54 and a maximum negative Moran's I index of −0.28. The residuals of the CAR model had a high similarity to OLS in terms of positive spatial autocorrelation, but the negative spatial autocorrelation was much weaker than OLS. Compared to the OLS and CAR models. The residuals of the SAR model show lower spatial autocorrelation, and although it shows some positive correlation up to 600 km (the highest positive correlation Moran's I index is 0.33), the remaining distance shows less spatial autocorrelation, and the SAR model residuals show the greatest spatial heterogeneity (Figures 5 and S4).

The relative importance of ART, MTC and MAP in explaining the endemic richness pattern of genus *Viburnum* was high. The explanatory variables varied in the interpretation of endemic richness patterns in the seven sections of the genus *Viburnum*. The PS and TS were the best interpreters of the richness patterns in Sect. *Viburnum*; PS and PDM were the best interpreters of the richness patterns of Sect. *Pseudotinus*; ART and MTC being the best interpreters of the richness patterns of Sect. *Tinus*. TS and PS were the best interpretations of the richness pattern of Sect. *Solenotinus*; MAP and PDM were the best interpretations of the richness pattern of Sect. *Tomentosa*; ART and MT were the best interpretations of the richness pattern of Sect. *Megalotinus*; PDM and PS were the best interpretations of the richness pattern of Sect. *Odontotinus* (Figures 6 and S5).

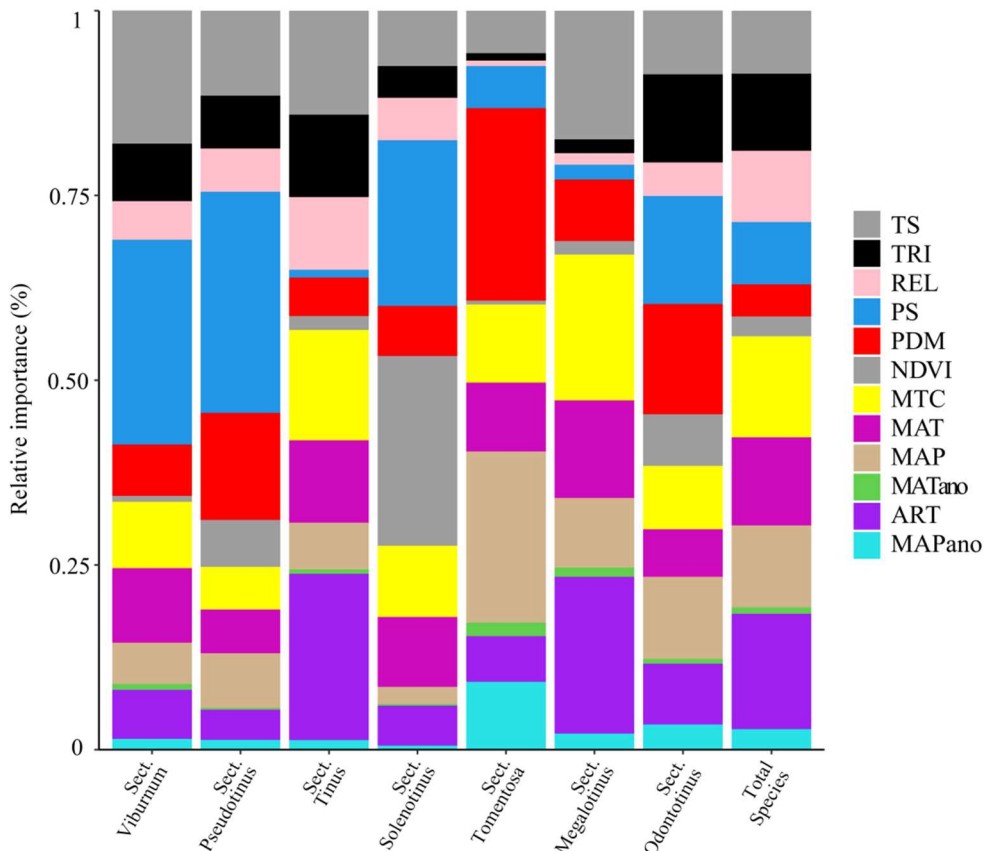

**Figure 6.** The relative importance of each explanatory factors for the richness of seven Sections of endemic taxa of genus *Viburnum* in China.

## 4. Discussion

Endemic members of genus *Viburnum* in China exhibit a very high richness, accounting for about 62% of all species (including inner species taxa) of the genus in China. They are widely distributed in China and are absent only in a few northern provinces. According to our results, China is undoubtedly an important distribution and endemic center for the genus *Viburnum* in Asia. However, the centers show inconsistent results at different scales. When assessed at the provincial scale, the endemic centers are in Yunnan and Sichuan provinces. When analyzed by raster (50 km × 50 km) processing, endemism centers were found to be mainly in the mountainous regions of southern China, such as the Qionglai-Wumeng Mts., the Qinling-Daba Mt, the Wuling Mts., the Nanling Mts., the Wuyi Mts. and the Central Mts. of Taiwan. Compared to administrative units, the rasterized method provides a clearer description of species distribution patterns and is more conducive to species conservation studies.

Our study also supports the climate seasonality hypothesis, with climate seasonality being the strongest explanation for the richness pattern of endemic taxa of *Viburnum* in China. The same findings have been reported in explanatory studies of the plant richness patterns in China for family Gesneriaceae, Moraceae, genus Rhododendron among others [4,51,52]. There is large seasonal differences in climate between the north and south of China. For example, the extreme temperature in winter, when the fluctuations are most pronounced, are higher in the northern regions than in the southern regions [53]. Such extremes temperatures may be an important reason for the restricted distribution of genus *Viburnum* in northern China, while the moderate climatic seasonality of southern and southwestern China creates stable climatic conditions for the survival and spread of taxa of this genus endemics [54], which supports the tropical niche conservatism theory (most lineages originated in tropical climates, and that they colonized extratropical and

temperate areas more recently) [55,56]. Meanwhile, since many taxa of *Viburnum* seeds have dormant properties, such as *V. plicatum* var. *formosanum*; *V. sargentii*; *V. betulifolium* and *V. parvifolium* [18,57,58]. The seasonality of the climate (especially the low temperature and its duration) may promote the de-dormancy of the seeds, which will also affect the distribution pattern of endemic species of *Viburnum*.

Habitat heterogeneity and historical climatic changes have a relatively weak influence on the geographic pattern of endemic taxa richness of *Viburnum* compared to the influence of water-energy synergistic effects. However, it is undeniable that mountainous areas with high habitat heterogeneity are the regions with the highest endemic taxa richness of *Viburnum* spp. These areas of high habitat heterogeneity are mainly due to altitudinal differentiation and have a wide range of habitats with diverse hydrothermal conditions [53,59]. Many studies have demonstrated that habitats with high heterogeneity can promote the coexistence and differentiation of multiple species, and also limit the dispersal of some species which are more dependent on specific habitats [9,35,60]. These mountains also act as refuges for species in harsh climatic conditions, for example, the last glaciation affected species richness through habitat isolation and limited migrations [61]. Moreover, the highly heterogeneous tropical and subtropical mountain ranges in China have been proved to be the cradle of the Chinese flora, thus dominating the species richness and concentrations of narrow endemic species [53,61–63].

The geographic patterns and distribution centers of different sections of endemic taxa of the genus *Viburnum* also show inconsistencies. Such divergent patterns can facilitate the understanding of species evolution and biogeography of genus *Viburnum* in China. According to the distribution pattern of the seven sections obtained in this study, and combined with the phylogenetic tree of *Viburnum* constructed by Ran [23], we speculate that the temperate forest in Southwest China may be an important differentiation center of *Viburnum*, which gradually spread and evolved to the eastern and southeastern regions of China. This may be the reason for the frequent niche shifts (especially the multiple and multi-directional migration between the colder deciduous temperate forest and the warmer evergreen temperate forest) and the multiple evolutionary shifts of the genus *Viburnum* since the Oligocene have been responsible for the species diversification [31,35], while the southwest mountainous area with diverse terrain provides a natural place for species preservation and differentiation. In the process of its eastward and southeastern migration, differences in the geographical pattern of the current differentiation centers of the seven sections were gradually formed due to the geographical constraints of the mountains as well as climatic differences. However, more evidence is needed to prove the geographical distribution and evolutionary history of *Viburnum* in China.

To date, nine endemic taxa of *Viburnum* have been listed on the IUCN categories in China. These species have been categorized as: Vulnerable (2 species), Near Threatened (3 species), Endangered (2 species) and one species as Critically Endangered (Appendix). Narrow distribution, small populations and over-harvesting for their high ornamental value are the main reasons for the endangerment of these endemic species. Actually, in addition to these species already listed on the IUCN categories, there are many other endemic species of the genus *Viburnum* in China that are also threatened by such negative factors. In particular, the 24 endemic and rare taxa of the genus *Viburnum* that we have collated in this study. For example, *V. omeiense* and *V. tengyuehense* var. *polyneurum* have only type specimen records and no further wild populations have been reported in past recent decades, and are most likely extinct in the wild. The *V. squamulosum*, *V. fengyangshanense* and *V. corymbiflorum* subsp. *malifolium* have an extremely narrow distribution range and extremely sparse populations in the wild. The *V. leiocarpum* var. *punctatum* and *V. triplinerve* are less common in the wild due to the severe destruction of their habitat. Despite the fact that many mountains in China have been classified as protected areas, which greatly contributing to the conservation of plant diversity. However, the coverage of protected areas is still incomplete and discontinuous in the distribution hotspots of endemic and rare species of *Viburnum* obtained in this study (Figure 3H). What's more, human activities have

significantly altered the landscape and will also inhibit the spread of many species. Thus, endemic taxa of *Viburnum* face serious challenges in the future of global changes. Here, we hereby call on conservationist to focus on the following issues in the future to promote the conservation of *Viburnum* spp. in China: (a) strengthening field distribution and population size surveys and reassessing the IUCN categories of all its taxa; (b) strengthen research on the ecological adaptation mechanisms and reproductive biology of those species with narrow distributions; (c) strengthen research into the causes of the endangerment and rarity of some taxa; (d) increase the artificial propagation of endangered and highly ornamental species to promote the conservation of wild populations; (e) the impact of climate change on the distribution patterns of endemic species and how to cope with it.

**Supplementary Materials:** The following supporting information can be downloaded at: https://www.mdpi.com/article/10.3390/d14090744/s1, Figure S1: Frequency distribution of endemic richness of genus *Viburnum* in China within the 50 × 50 km grids; Figure S2. Correlation between the endemic richness of genus *Viburnum* and longitude (A) and latitude (B) in China; Figure S3. Linear relationships between explanatory variables and endemic richness of genus *Viburnum*; Figure S4. Geographic distribution of residuals for the three model tests for endemic richness of genus *Viburnum* in China; Figure S5. Magnitude of variation in each of the explanatory factors for the richness of seven Sections of endemic of genus *Viburnum* in China; Table S1. Checklist of endemic species of the genus *Viburnum* in China, with their endangered status, and AUC values for species distribution model results; Table S2. Spearman's rank correlation coefficients for 8 bio-climate predictors.

**Author Contributions:** Conceptualization, S.W. and W.L.; data curation, W.L. and J.Y.; formal analysis, S.W., S.D., W.L. and J.Y.; funding acquisition, S.W. and H.L.; investigation, W.L., S.H. and J.Y.; methodology, S.W.; resources, W.L.; software, S.D.; supervision, H.L.; writing—original draft, S.W.; Writing—review and editing, S.W., V.M.N. and W.L. All authors have read and agreed to the published version of the manuscript.

**Funding:** This work was supported by the National Natural Science Foundation of China (32100166) and the Special Research Assistant Program of Chinese Academy of Sciences.

**Institutional Review Board Statement:** Not applicable.

**Informed Consent Statement:** Not applicable.

**Data Availability Statement:** Not applicable.

**Acknowledgments:** We thank Ming Tang from Jiangxi Agricultural University for helping to provide distribution information for *V. aquamulosum*. Thanks to Qiang Fu from Huazhong Agricultural University, Boshun Xia from Wuhan Botanical Garden, Chinese Academy of Sciense, Xiaoling Chen from Lijiang Aline Botanic Garden, Renkun Li from Enshi Dongsheng Plant Development Co., Ltd. for helping to Provide photo 6, 28, 40, 46 of Figure 1.

**Conflicts of Interest:** The authors declare no conflict of interest.

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
