# Peer review of "Spatial Patterns and Determinants of Endemic Taxa Richness in the Genus Viburnum (Adoxaceae) in China"

_diversity, doi:10.3390/d14090744_

Round 1

Reviewer 1 Report

Dear editor:

   I finished reviewing the manuscript named “Spatial patterns and determinants of endemic taxa richness in the genus Viburnum (Adoxaceae) in China”. The genus Viburnum is widely distributed in China and has great ornamental value. In this manuscript, the authors determined the distribution information of 61 endemic taxa of Viburnum from specimens’ review and field surveys, as well as species distribution model. This study has great practical significance. I have raised some questions about the ecological significance here. I hope to get clarification from the authors.

Major issues:

1. The authors carefully summarize the literatures of Viburnum in the introduction. Before the last paragraph of the introduction, I would like to see a paragraph summarizing some research about the diversity, biogeography, ecological habits and even pollination of Viburnum (maybe not for the whole genus but some species of this genus) in recent years. And draw a conclusion here, there is no such study focus on the diversity pattern of the whole genus.

2. Is it feasible for the authors to use climate and environmental data to explain the species richness simulated based on climate data?

3. I think it is more ecologically meaningful to study the whole genus rather than just endemic ones.

4. I do not understand the ecological significance of dividing the genus Viburnum into seven sections and use climate varibales to explain the diversity patterns of each sections? The seven sections are only systematic relationships, not climate adaptation. This needs to be clarified by the authors, especially in the introduction section. Could the authors consider a more appropriate grouping method?

5. As I mentioned above, the authors did not discuss the differences in climate interpretation of the seven sections in the discussion part.

Minor issues:

Line 26: The seven sections mentioned here is not clear.

Line 29: This sentence has no logical relationship with the previous text. Or the reader does not know the relationship between Yunnan Guizhou Plateau, Sichuan Basin and Yunnan Province, so there is no causal relationship here.

Line 29: What is “Environmental energy”?

Line 39-40: What is “heterogeneous distribution”? I think biodiversity is unevenly distributed because of the environment and climate. And, maybe this topic is not been “debate among ecologists”, this is only a hot research field.

Line 45: There are many hypotheses to explain the patterns of species richness, but I think not as much as hundreds.

Line 61: I don't think ecological theory needs to be verified in specific ornamental plants. Or, don't mention ornamental here. Although the Viburnum is ornamental, it is not suitable to be mentioned here.

Lines 71-72 and Lines 74-76: I think you repeated the same meaning in this paragraph, i.e., diversity pattern of Viburnum in China is not clear.

Line 101: What is “distribution raster”? And what is the relationship between 100 raster and 25 point?

Lines 122-123: At least it should not be introduced “raster” here. It should be placed before Line 101.

Liens 185-189: Is the 4162 grid cells with Viburnum sampled from species distribution model (SDM)? And “the 1764 grid cells had recorded”, this data is recorded from specimens collections? What data set was used in analysis, please clarify.

Lines 306-307: I don't think we should raise such a question in the first paragraph of the discussion. This is not the scope of this article. Can you put it in the last paragraph to extend this study?

Reviewer 2 Report

The manuscript entitled “Spatial patterns and determinants of endemic taxa richness in the genus Viburnum (Adoxaceae) in China” reported a study on distribution patterns and formation mechanisms of the endemic taxa of Viburnum and their seven sections, which will be useful for developing scientific conservation and appropriate management strategies for endemic taxa in China. All aspects of the study are carefully designed, the data were analyzed thoroughly, the results were discussed thoroughly, and the figures and tables were also well presented. However, it is better to polish the language carefully to avoid mistakes in grammar and typos.

I only have a few minor comments that may help to improve the quality of the manuscript. 

1. Line 31. Perhaps, the sentence ‘However, the explanation of environmental factors … Viburnum’ could be improved.

2. Line 39. ‘…its formation mechanism…’ should be ‘…their formation mechanisms…’.

3. Line 44. Perhaps ‘the best sites’ could be replaced with ‘priority sites’.

4. Lines 47-49. The relationship between ‘Particularly … at large scales’ and the following paragraph is not clear. I recommend to rephrase this sentence. 

5. Line 63. ‘Viburnum. L’ should be ‘Viburnum L.’

6. Lines 65-67. Singular and plural nouns should be used correctly. Please check.

7. Line 70. add ‘the’ before ‘genus Viburnum’.

8. Line 73. ‘as’ should be ‘is’?

9. Lines 78-80. I recommend to use declarative sentences. e.g., ‘to 1) examine the geographic distribution pattern …, and 2) reveal the possible causes …’.

10. Line 99. ‘remove’ should be ‘removing’

11. Line 101. For ease of reading and understanding, it is better to give an explanation of ‘a distribution raster count’. 

12. Line 102-103. Is there any unit after the number?

13. Line 115. Please describe the time period for nineteen bioclimatic variables used.

14. Line 143. ‘Viburnum’ should be italicized.

15. Line 149. Perhaps ‘…spatial autoregressive models are the conditional autoregressive…were chosen…’ should be …spatial autoregressive models, the conditional autoregressive…were chosen…’ (‘are’ should be replaced by ‘,’)

16. Lines 153-154. The sentence ‘… and the value of AIC …other models’ is not clear. Please rephrase.

17. Line 156. add ‘further’ before ‘explore’?

18. Lines 166-170. the sentence ‘Sect. Viburnum … sect. Tomentosa 2 taxa’ seems to be grammatically incorrect. Please rephrase. Besides, should the scientific name of section be italicized? Please check.

19. Line 186. Perhaps, ‘occurrence species’ should be ‘species occurrence’.

20. Line 190. add ‘the’ before ‘genus Viburnum’.

21. Line 192. The phrase ‘… most specimen collection mainly recorded…’ would be less redundant as ‘…specimen records mostly from …’? Please check.

22. Lines 195-198. Maybe this sentence could be improved. 

23. Lines 200-202. The sentence ‘we found that … Mountains’ seems to be grammatically incorrect. Please rephrase.

24. Line 203-204. ‘…showed higher richness.’ should be ‘…showed higher richness than …’ or ‘Comparing to …, … showed higher richness’.

25. Line 224 and Line 228. ‘Province’ should be ‘Provinces’.

26. Lines 223-224. Maybe this sentence could be improved. 

27. Lines 234-236. The extensive use of abbreviations makes the text difficult to read and understand. I recommend that the authors provide the full explanation for each abbreviation on its first occurrence in each section (M&M, Results and Discussion).

28. Lines 317-318. ‘Thus, our results generally supported the environmental energy hypothesis’ seems more reasonable.

29. Line 321. ‘Viburnum’ should be italicized.

30. Line 337. Perhaps, it is better to make a brief introduction to ‘the tropical conservatism theory’.

31. Line 344. Perhaps, ‘the water-energy hypothesis’ should be ‘the influence of water-energy’.

32. Line 356. Why use ‘only’? Besides, ‘Viburnum’ should be italicized.

33. Line 363. ‘Viburnum’ should be italicized.

Reviewer 3 Report

The article “Spatial patterns and determinants of endemic taxa richness in the genus Viburnum (Adoxaceae) in China” seeks to answer the question of evolution and isolation of endemic species of the genus Viburnum in China, even presenting a proposal for the center of genetic origin. So it is suitable for publication in the journal Diversity.

In general, the article is well organized and has a good quality. However, the authors need to clarify some issues.

The Abstract is well prepared and focuses on the main ideas developed in the article.

The Introduction adequately justifies and highlights the importance of carrying out this study. However, it is necessary to be careful because in Europe there are also at least 3 taxa of the genus Viburnum. Authors should not generalize the distribution area of the genus Viburnum.

Regarding the methodology, it seems to me to be adequate for the type of analysis intended. However, the authors should mention the taxonomic nomenclature adopted. In addition, each botanical name must be accompanied with the corresponding classifier. I understand that the authors do not use it throughout the text, but at least the table in Appendix A must be complete. Missing classifier name: V. macrocephalum f. macrocephalum, V. chingii var. chingii, V. corymbiflorum subsp. corymbiflorum, V. odoratissimum var. arboricola, V. tengyuehense var. tengyuehense, V. leiocarpum var. leiocarpum, V. formosanum var. formosanum, V. sempervirens var. sempervirens.

The genus Viburnum is thought to have possibly emerged during the Paleogene Period (https://doi.org/10.1007/978-94-009-9987-9_20), when the planet's climate was warmer and wetter. However, the authors refer several times to mountain areas (lower temperatures). The authors should clarify the altitude of occurrence of the various taxa of the genus Viburnum, because the reader can understand that most plants like the cold and this is not entirely true.

Isn't the fact that several threatened taxa of the genus Viburnum have a high ornamental interest a positive thing? In order to increase the number of individuals of the endangered species, the authors can propose their propagation in nurseries, in order to be used in urban parks and gardens. The creation of a living gene bank could prevent the extinction of some taxa, as mentioned by these authors:  DOI: 10.31586/rjees.2022.

I think this important article has not been seen by the authors: https://doi.org/10.1086/658927234

Round 2

Reviewer 3 Report

I really enjoyed seeing the information added in detail to the article. The authors are to be congratulated for the good work done. However, it is still necessary to clarify the taxonomic authorship of some plants. I advise authors to follow a broader taxonomic nomenclature, for example the International Plant Name Index (https://www.ipni.org). All names have a classifier. If it is not on this list, it should be searched for in specialty authors and more recently published articles.

As I am satisfied with the answers presented, I consider that the article can be published after introducing the missing classifiers.

Author Response

 Thank you very much for your comments and suggestions. The names of all taxa in the Appendix have been corrected with classifier name according to the Reviewer’s comments. However, because of the classification of Viburnum in China remains questionable, and it’s under constant revision. Therefore, not all classifier of botanical name in this study were accompanied according to IPNI, some according to the recently published articles and FOC.

In the future, We will also conduct research on the taxonomic revision of Viburnum in China,the study will be published in my next paper, so stay tuned.